# Effects of Rumen-Protected β-Alanine on Growth Performance, Rumen Microbiome, and Serum Metabolome of Beef Cattle

**DOI:** 10.3390/ani16010043

**Published:** 2025-12-24

**Authors:** Daci Fu, Kang Mao, Yihao Zang, Mingren Qu, Qinghua Qiu, Xianghui Zhao, Kehui Ouyang, Yanjiao Li

**Affiliations:** Jiangxi Province Key Laboratory of Animal Nutrition, Animal Nutrition and Feed Safety Innovation Team, College of Animal Science and Technology, Jiangxi Agricultural University, Nanchang 330045, China; 17346713930@163.com (D.F.); kang3681488@163.com (K.M.); zangyihao@126.com (Y.Z.); mingrenqu@jxau.edu.cn (M.Q.); rcauqqh@cau.edu.cn (Q.Q.); xianghuizhao@jxau.edu.cn (X.Z.); ouyangkehui@sina.com (K.O.)

**Keywords:** beef cattle, rumen-protected β-alanine, growth performance, rumen fermentation, rumen metabolomics

## Abstract

This study selected Simmental crossbred cattle to investigate the impact of rumen-protected β-alanine (RP-β-Ala) on growth performance and its underlying mechanisms. The findings revealed that RP-β-Ala improved average daily gain and crude protein digestibility and reduced the gain-to-feed ratio. Additionally, RP-β-Ala also reshaped the rumen microbiota, enriching beneficial genera, including *Prevotella*, *Treponema*, and *Selenomonas*, which increased volatile fatty acid production and nitrogen utilization efficiency. Metabolomics show alterations in arachidonic acid metabolism and enhanced antioxidative capability. RP-β-Ala reduced serum TNF-α levels by suppressing harmful bacteria and inhibiting polyamine synthesis, alleviating oxidative stress and inflammation. Overall, RP-β-Ala enhances beef cattle growth through improved energy supply and antioxidant capacity.

## 1. Introduction

With the increasing demands for beef quality and quantity, optimizing the feeding management of beef cattle and enhancing their growth performance have become hot topics for research priorities. The intake of protein nutrition is one of the key factors determining the growth of beef cattle [1]. However, when the crude protein content in the diet exceeds the requirements of rumen microbes, it has dual adverse effects: reducing nitrogen utilization efficiency and increasing emissions of urinary nitrogen and methane, which contribute to environmental pollution [2,3]. Therefore, improving the host’s nitrogen utilization rate can not only reduce feed costs but also enhance the growth performance of the host [4]. Supplementing amino acids in the diet has been proven to be a practical and effective method. For example, Li et al. [5] found that the growth performance of yaks was significantly improved by supplementing rumen-protected (RP) lysine. In addition, supplementing ruminants with RP methionine [6] and RP leucine [7] also shows positive promoting effects.

β-Ala, an isomer of alanine identified as 3-aminopropionic acid, exhibits a molecular mass of 89.09 and composition C_3_H_7_NO_2_ [8]. Although β-Ala is a non-essential amino acid, it has been proven to promote muscle development and improve meat quality in various animal models [9]. Studies have shown that β-Ala supplementation can improve feed conversion and growth performance in broiler chickens [10] and enhance meat quality [11,12]. These responses could be attributed to the host’s improved redox balance and upregulated energy metabolism. Alanine is decarboxylated to produce pyruvate, which enters the tricarboxylic acid cycle to provide energy for the body [13]. Apart from being a carnosine precursor, β-Ala is also a constituent of pantothenic acid (vitamin B_5_), which is an essential component of coenzyme A (CoA). CoA acts as the universal carrier of acyl groups, thereby directly participating in the catabolic and anabolic pathways of carbohydrates, lipids, and amino acids [14]. Additionally, β-Ala can increase the content of carnosine in muscles, thereby enhancing the body’s antioxidant capacity and having a certain degree of safety [15]. Overall, β-Ala has important application value in animal production and sports nutrition.

However, research on β-Ala in the field of ruminants is still relatively limited. Zhao et al. [16,17] reported that irrespective of β-Ala being rumen-protected, it can significantly improve the nitrogen utilization efficiency of beef cattle. Specifically, the direct supplementation of β-Ala can improve nitrogen utilization efficiency by regulating rumen microbes (mainly by increasing the abundance of *Prevotella*), but it has no significant effect on growth performance. Under rumen-protected conditions (from ruminal degradation to release in the small intestine), β-Ala can improve nitrogen utilization efficiency by reducing the concentration of taurine in plasma. Although these studies have shown that β-Ala has positive effects, there are some limitations. On the one hand, the small number of experimental animals may not accurately reflect the authenticity and reliability of the data. A study used a 3 × 3 Latin-square design conducted in six beef steers (391 ± 10 kg BW) with three 20 d periods (15 d adaptation + 5 d sampling) and three RP-β-alanine levels (0, 17.5, and 35 g d^−1^) and found insignificant improvement of growth performance. On the other hand, there is still a lack of sufficient data to support the effect of RP-β-Ala on growth performance. β-alanine supplementation consistently improves broiler performance [10]; published ruminant studies have reported no significant increases in body weight gain [16]. Although the main purpose of rumen-protected amino acids is to protect their absorption and utilization in the small intestine, their presence in the rumen may still indirectly affect rumen microbes by changing the rumen metabolic environment and regulating the balance of microbial communities [5,18].

Therefore, the purpose of this study is to examine how RP-β-Ala influences the growth performance of beef cattle by adding it to the diet and to explore its specific mechanisms of action by analyzing rumen microbes and their metabolites. This study is intended to provide a new perspective and scientific basis for the application of β-Ala in ruminant nutrition.

## 2. Materials and Methods

### 2.1. Animal Treatment and Experimental Diets

The animal protocol received ethical clearance from Jiangxi Agricultural University’s Laboratory Animal Welfare Committee (approval No. JXAULL-2024-03-34). Thirty-six 18-month-old Chinese Simmental crossbred bulls with similar weight (627 ± 41 kg) were randomly assigned to 1 of 2 treatments (Con and experimental groups); each group contained six replicates (pens) with three cattle each. A total of cattle were divided into two groups using a randomized design. While the control group received only the basal diet, the treatment group was administered an additional 96 g/d/cattle RP-β-Ala. β-Ala, with a purity of 99.9%, was obtained from Shandong Jiaoyang Biotechnology Co., Ltd., located in Jining, China. The production of fat-encapsulated RP-β-Ala was contracted to Hangzhou King Techina Feed Co., Ltd., based in Hangzhou, China. The coating rate for β-Ala was determined to be 50%, the rumen bypass ratio after 12 h was found to be 86.3%, and the dissolution in intestinal fluid after 12 h was recorded at 81.7%. In this study, the supplemental dosage of RP-β-Ala was based on the findings of Hu et al. [17]. RP-β-Ala was incorporated into the concentrate and administered twice daily at 07:00 and 19:00 h, while all animals were provided with feed and water ad libitum. The 28-day feeding experiment was conducted using a basal diet whose ingredient and nutrient profiles are detailed in Table 1.

### 2.2. Growth Performance Measurement and Sample Collection

The weights of beef cattle were assessed with an electronic livestock scale (TGT-500, Shanghai Yaohua Weighing System Co., Ltd., Shanghai, China), and the feed consumption was documented over three consecutive days at intervals of seven days. This information was utilized to determine growth metrics such as average daily gain (ADG), average daily feed intake (ADFI), and feed-to-gain ratio (F/G). Fecal samples were collected via rectal palpation 72 h prior to the conclusion of the study, concurrent with the collection of dietary samples [19]. All fecal specimens were thoroughly mixed, with 10% H_2_SO_4_ added to fix nitrogen (Sinopharm Chemical Reagent Co. Ltd., Shanghai, China). Both feed and fecal samples were preserved at −20 °C until the time of analysis. At the conclusion of the experiment, a 10 mL blood sample was drawn from the jugular vein of the cattle into evacuated tubes without anticoagulant tubes. The blood samples were subjected to centrifugation at 3000× *g* for 10 min at 4 °C (3–18 K, Sigma Laborzentrifugen GmbH, Osterode am Harz, Germany) to separate serum. Following the collection of blood samples, rumen fluid was gathered using an oral stomach tube (120 cm × 10 mm i.d., custom-made, local veterinary supply, Zhengzhou, Henan, China). During the sampling intervals, the tube was rinsed with warm, clean water; the first rumen fluid was discarded to prevent contamination by saliva. A new 50 mL aliquot was then obtained, and its pH was immediately measured using a portable meter (PHS-3C, Shanghai Leizi, Shanghai, China), divided into sterile tubes, snap-frozen in liquid nitrogen, and stored at −80 °C until further analysis.

### 2.3. Chemical Analyses

Samples of feed and feces were subjected to drying in an electrically heated oven (DHG-9070A, Yiheng, Shanghai, China) at a temperature of 65 °C for a duration of 48 h and then ground using a 1 mm stainless steel mesh (Standard laboratory sieve, local machine shop, Nanchang, China). Following the methodology outlined by Van et al. [20], analysis was conducted on the content of DM, CP, ether extract (EE), acid detergent fiber (ADF), and neutral detergent fiber (NDF) in samples from both feed and manure. For the test cattle, the apparent digestibility of nutrients was determined utilizing the acid-insoluble ash (AIA) method [21]. The concentration of ruminal ammonia nitrogen (NH_3_-N) was assessed using the colorimetric method as detailed by Li et al. [22]. All reagents—phenol, sodium hypochlorite (≥5% available chlorine), and sodium hydroxide—were sourced from GuanYin Chemical Reagent Co. Ltd., Nanchang, China. The determination of ruminal MCP was carried out utilizing the colorimetric method outlined by Mao et al. [23]. This process involved the use of a high-speed refrigerated centrifuge (3–18 K, Sigma Laborzentrifugen GmbH, Osterode am Harz, Germany), which functioned at a force of 12,000× *g* for a duration of 15 min. Subsequently, absorbance was measured at 595 nm with the aid of a UV–Vis spectrophotometer (T6 New Century, Beijing Purkinje General Instrument Co. Ltd., Beijing, China). The rumen fluid’s VFA concentrations were measured using gas chromatography (GC-2014, Shimadzu, Kyoto, Japan), following the procedure outlined by Li et al. [24]. The serum concentrations for levels of total protein (TP), total cholesterol (TC), triglycerides (TG), and blood urea nitrogen (BUN) were conducted using a fully automated biochemical analyzer (BS-420, Mindray Bio-Medical Electronics Co., Ltd., Shenzhen, Guangdong, China), in accordance with the manufacturer’s guidelines provided by the relevant supplier (BioSino Bio-Technology and Science Inc., Beijing, China). Serum levels of TNF-α, IL-1β, and IL-4 were measured using ELISA kits from the Beijing Sino-UK Institute, whereas the antioxidant markers—SOD, GSH-Px, T-AOC, and MDA—were assessed with commercial assays provided by Nanjing Jiancheng in China.

### 2.4. 16S rDNA Sequencing Analysis

DNA was isolated from a sample of rumen fluid utilizing the E.Z.N.A.^®^ soil DNA kit (Omega Bio-tek, Norcross, GA, USA). The V3-V4 16S rRNA genes were amplified through PCR using the primers 338F (5′-ACTCCTACGGGGAGGCAGCAG-3′) and 806R (5′-GGACTACHVGGGTWTCTAAT-3′). Amplicons were cleaned with the AxyPrep gel extraction kit (Axygen, Union City, CA, USA), quantified on a Quantus™ Fluorometer (Promega, Madison, WI, USA), and sequenced on Illumina MiSeq PE300 (Illumina, San Diego, CA, USA). Quality trimming was performed with fastp v0.19.6 [25]. Raw data have been uploaded to NCBI under BioProject PRJNA1208407. Paired-end reads were merged with FLASH v1.2.11 [26], quality-filtered, and clustered into 97%-similarity OTUs by UPARSE v7.1 [27]. Taxonomy was assigned at ≥70% confidence via the RDP classifier v2.11 [28], and downstream analyses were executed on the Majorbio cloud pipeline. Alpha-diversity (Chao1, Shannon, Simpson, Faith_pd, etc.) and beta-diversity metrics were computed in QIIME 2; group-wise differences in alpha-diversity were tested with the Wilcoxon rank-sum test (ggpubr v0.6.2) in our R 4.4.1 session [29]. PCoA ordinations were visualized through Emperor, community bar plots were drawn in R v4.4.1, and indicator genera (LDA score > 2) were identified with LEfSe.

### 2.5. Metabolomics Analysis by Liquid Chromatography–Mass Spectrometry

Rumen metabolites were extracted as described [22] and profiled on an LC-MS system. Raw files were imported into Progenesis QI for peak picking and alignment, generating a matrix of retention time, *m*/*z*, and intensity. Features present in <80% of any group were dropped, missing values were imputed with the minimum detected value, and the data were sum-normalized. QC peaks showing RSD > 30% were removed, the matrix was log10-transformed, and metabolites were annotated against HMDB and METLIN. The processed data were submitted to the Majorbio platform, where orthogonal partial least squares discriminant analysis (OPLS-DA) and principal component analysis (PCA) were employed. Metabolites identified as significantly different had a variable importance in projection (VIP) exceeding 1 and a *p*-value less than 0.05, according to the findings from the OPLS-DA model and the Student *t*-test *p*-values. For the identification of pathways related to differential metabolites, metabolic pathway annotation was conducted using the Kyoto Encyclopedia of Genes and Genomes (KEGG) database (http://www.genome.jp/kegg/, accessed on 10 December 2023)

### 2.6. Associations Between Species and Pathways

The relationships between various bacterial species, DEMs, and serum phenotypic indicators were evaluated through Spearman’s rank correlation across all samples. Additionally, the connections among bacterial species, DEMs, and phenotypic indices were visualized via a heatmap utilizing the R package ‘pheatmap’ (version 1.0.13) [30].

### 2.7. Statistics and Data Analysis

Growth performance, blood metabolites, and rumen fermentation parameters were analyzed with SPSS 20.0 (IBM, Chicago, IL, USA). The model was Yij = μ + Treatmenti + εij. Normality was checked by Shapiro–Wilk; normally distributed variables were compared between Con and RP-β-Ala using an independent-samples *t*-test, while non-normal data were analyzed with the Mann–Whitney U test. For 16S rRNA, differences between the two groups were tested with the Kruskal–Wallis rank-sum test, and pairwise comparisons were performed using Dunn’s test with FDR correction (tendency 0.05 ≤ *p* < 0.10, * *p* < 0.05, ** *p* < 0.01) in R 4.4.1 (vegan, rstatix).

## 3. Results

### 3.1. Growth Performance and Apparent Digestibility of Nutrients

As illustrated in Figure 1, dietary supplementation with RP-β-Ala led to a tendency enhancement in ADG (*p* = 0.065) and a reduction in the F/G ratio (*p* = 0.078) in beef cattle when compared to the Con group. However, there were no notable differences in ADFI between the two groups. Regarding the apparent digestibility of nutrients, there were no significant differences in the levels of EE, NDF, ADF, and DM (*p* > 0.05) between the two groups. Conversely, the RP-β-Ala group exhibited greater CP digestibility relative to the Con group (*p* = 0.065).

### 3.2. Serum Indices

As illustrated in Figure 2, the RP-β-Ala group notably decreased serum levels of BUN (*p* = 0.037) and TNF-α (*p* = 0.01) when compared to the Con group while also enhancing T-AOC levels (*p* = 0.025). Conversely, no significant differences were observed in serum TP, TC, TG, SOD, GSH-Px, IL-1β, and IL-4 between the two groups (*p* > 0.05).

### 3.3. Rumen Fermentation Indicators

As shown in Figure 3, the group that received dietary supplementation with RP-β-Ala exhibited a notable decrease in the ruminal pH value (*p* = 0.01) compared to the Con group. Moreover, significant increases were noted in the ruminal NH_3_-N levels (*p* = 0.006) and MCP levels (*p* = 0.025) for the RP-β-Ala group. The total VFA (TVFA) levels (*p* = 0.016), along with acetate (*p* = 0.016) and propionate (*p* = 0.01), were significantly higher in the RP-β-Ala group in contrast to the Con group. Additionally, a trend towards a lower acetate-to-propionate ratio was identified in the RP-β-Ala group compared to the Con group, although the butyrate concentration did not display any significant differences between the two groups (*p* > 0.05).

### 3.4. Rumen Bacterial Diversity

To investigate the changes in ruminal fluid microbiota between the Con and RP-β-Ala groups, 16S rRNA sequencing was employed to analyze the microbial composition of 12 ruminal fluid samples of beef cattle. A total of 8,794,453 readings were detected from the 12 samples, with an average of 732,871 per sample (Appendix A). The sequences were grouped into 14,058 OTUs, showing a 97% similarity. Within this grouping, 7025 OTUs were shared, while 2794 OTUs were exclusive to the Con group, and 4239 OTUs were unique to the RP-β-Ala group (Figure 4A). The α-diversity of the rumen microbiota for both groups was assessed using the Ace, Chao1, Shannon, Simpson, and Coverage indices, which reflect aspects of microbial diversity, richness, and evenness. The results showed that only the Shannon index showed a significant change (*p* < 0.05), indicating certain differences in microbial species richness and evenness between the two groups (Figure 4B). Further β-diversity analysis was performed to assess the differences between groups. The results showed significant differences in the phylogenetic distance of bacterial community structures between the Con and RP-β-Ala groups. PCA and PCoA analyses revealed that the Con and RP-β-Ala groups could be distinguished from each other, indicating that RP-β-Ala supplementation significantly altered the overall microbial diversity in the rumen of beef cattle (PERMANOVA *p* < 0.01). Additionally, the NMDS analysis yielded consistent results (stress = 0.05) (Figure 4C–E).

In parallel, we carried out further exploration to understand the variability in microbial composition across phylum, class, and genus levels. At the phylum level, the three predominant taxa observed were *Firmicutes*, *Bacteroidota*, and *Patescibacteria*. The leading five classes were *Clostridia*, *Bacteroidia*, *Negativicutes*, *Bacilli*, and *Saccharimonadia*. Similarly, the ten most prevalent genera were *Prevotella*, *Succiniclasticum*, *Rikenellaceae_RC9_gut_group*, *NK4A214_group*, *Christensenellaceae_R-7_group*, *Ruminococcus*, *Lachnospiraceae_NK3A20_group*, *Acetitomaculum*, *Candidatus_Saccharimonas*, and *Prevotellaceae_UCG-00* (Figure 4F–H). The LEfSe method with an LDA score > 2 identified the different microbial communities between groups. At the family taxonomic level, the RP-β-Ala group exhibited a rise in the prevalence of *Prevotellaceae*, *Vermiphilaceae*, and *Victivalaceae* compared to the Con group (Figure 4I,J). In terms of the meta-stats analysis conducted at both the genus and species levels, notable between-group differences were identified in the abundances of 36 genera and 28 species; specifically, the abundance of 20 genera and 7 species was greater than 0.01%. The differences in taxa abundance at both genus and species levels are illustrated as violin plots to provide a visualization of the data’s distribution characteristics (Appendix A). *g_Prevotella*, *g_UCG-004*, *g_Treponema*, *g_Selenomonas*, *g_Moryella*, *s_Selenomonas_ruminantium_subsp._lactilytica_TAM6421*, *s_Prevotella_ruminicola*, *s_Prevotella_sp._BP1-145*, *s_Selenomonas_ruminantium_AB3002*, *s_Prevotella_aff._ruminicola_Tc2-24*, and *s_Treponema_ruminis* were significantly enriched in the RP-β-Ala group compared to the Con group. Contrastingly, *g_Mogibacterium*, *g_Blautia*, *g_Lachnospiraceae_XPB1014_group*, *g_DNF00809*, *g_Desulfovibrio*, *s_Thermoactinomyces_vulgaris*, *s_Bacillus_coagulans_g_Bacillus*, *s_Laceyella_sacchari*, and *s_Acinetobacter_lwoffii* were lower in the RP-β-Ala group than in the Con group (Figure 4K,L).

### 3.5. Rumen Fluid Metabolomics

In order to investigate the primary metabolites associated with the Con and RP β Ala groups, twelve ruminal samples underwent analysis through non-targeted UPLC/MS metabolomics. Following a thorough quality screening and characterization process, a total of 1645 metabolites were identified from the rumen fluid of both groups. Of these, 1485 metabolites were annotated in the HMDB. The five superclass categories that exhibited the highest abundance were “lipids and lipid-like molecules (24.7%)”, “organic acids and derivatives (19.4%)”, “organoheterocyclic compounds (14.7%)”, “benzenoids (8.9%)”, and “organic oxygen compounds (8.0%)” (Figure 5A). Next, principal component analysis (PCA) was employed to assess the relationships among the different samples. The PCA visualizations indicated that the Con and RP-β-Ala groups demonstrated a high level of repeatability, whereas notable differences were found among the groups. Additionally, the results obtained from the partial least squares discriminant analysis (PLS-DA) model corroborated these findings (R2Y = 0.949, Q2 = 0.717) (Figure 5C,D). The analysis employing orthogonal partial least squares discriminant analysis (OPLS-DA) offered a comprehensive examination of the variances within the metabolite data, effectively distinguishing between the Con and RP β Ala groups (R2X = 0.434, R2Y = 0.955, Q2Y = 0.5706, as illustrated in Figure 5E,F). Following the *t*-test and VIP filtering based on the relative levels of rumen metabolites, a total of 191 differentially expressed metabolites (DEMs) were recognized between the two groups (with VIP > 1 and *p* < 0.05), comprising 136 metabolites that were upregulated and 55 that were downregulated in the RP β Ala group. Heat maps representing these DEMs in both groups are displayed in Figure 5G,H.

Furthermore, KEGG enrichment analysis of 191 identified metabolites revealed that they were mainly annotated to first-level metabolism categories: 4 DEMs were enriched in the arginine and proline metabolism (ko00330), 3 DEMs were enriched in the arachidonic acid metabolism (ko00590), 33 DEMs were enriched in the metabolic pathways (ko01100), 2 DEMs were enriched in the glutathione metabolism (ko00480), and 5 DEMs were enriched in the biosynthesis of cofactors (ko01240) (Figure 5I). In addition, the expression distribution of DEMs enriched in KEGG pathways was mapped between groups, as shown in Figure 5J and Appendix A.

### 3.6. Microbial Function Prediction Analysis and Correlation Analysis

To explore potential functional alterations associated with the rumen microbial changes in beef cattle, CAZy, KEGG, and MetaCyc module enrichment analysis was predicted by Picrust2 between the Con and RP-β-Ala groups (Figure 6A–C). There were no significant differences between the two groups on any of these factors. To understand the functional relevance of the microbial alterations for the growth performance of beef cattle, we analyzed the associations among the DEMs, serum phenotypic indicators, and differential bacterial species within each group. Bacteria from the species *Prevotella_sp._BP1-145*, *Proteus_mirabilis*, and *Treponema_saccharophilum_DSM_2985* were positively correlated with the metabolites 9-oxo-nonanoic acid (C16322), pretyrosine (C00826), 2-cis,4-trans-xanthoxin (C13453), 2-isopropyl-3-oxosuccinate (C04236), and 14,15-DiHETrE (C14775), and negatively correlated with the metabolites spermine (C00750) and uracil (C00106). Moreover, for the 26 upregulated DEMs in RP-β-Ala groups, *Thermoactinomyces_daqus* was negatively correlated with 5 DEMs, *Saccharopolyspora_rectivirgula_g__norank* was negatively correlated with 4 DEMs, and *Cerasibacillus_quisquiliarum* was negatively correlated with 3 DEMs (Figure 6D). In addition, it is worth noting that *Thermoactinomyces_daqus* and *Saccharopolyspora_rectivirgula_g__norank* were also negatively correlated with ruminal TVFA, acetate, and propionate (Figure 6E).

## 4. Discussion

### 4.1. Effect of Supplementary Rumen-Protected β-Ala on Growth Performance and Feed Conversion in Beef Cattle

This study investigated the impact of RP-β-Ala on beef cattle’s growth performance and nutrient digestibility. The findings indicate that incorporating RP-β-Ala enhances ADG, F/G, and CP digestibility in these animals. These findings suggest that RP-β-Ala may promote beef cattle growth by enhancing energy utilization efficiency and protein synthesis. Firstly, dietary β-alanine supplementation significantly elevates serum carnosine levels, which in turn enhances energy metabolism and consequently improves host growth performance [10,13]. This metabolic pathway is crucial for meeting the energy demands of growing animals. Secondly, RP-β-Ala can enhance nitrogen utilization efficiency in the rumen [16]. Enhanced nitrogen utilization efficiency can promote the synthesis of MCP in the rumen. This is consistent with the observed increase in MCP digestibility in this study. Improved nitrogen utilization in the rumen can lead to more efficient protein synthesis and better overall growth performance. Moreover, Zhu et al. [31] demonstrated a direct correlation between improved crude protein utilization and enhanced growth performance in animals. This supports the notion that RP-β-Ala’s effects on protein metabolism can translate into tangible benefits for beef cattle growth. Overall, these results highlight the potential of RP-β-Ala as a nutritional supplement to optimize beef cattle production.

### 4.2. Effect of Supplementary Rumen-Protected β-Ala on Serum Biochemical Parameters in Beef Cattle

Oxidative stress, characterized by an imbalance between the production of reactive oxygen species (ROS) and the body’s antioxidant defenses, can lead to tissue damage and trigger inflammatory responses [32]. The T-AOC reflects the combined efficacy of both enzymatic and non-enzymatic antioxidant defense systems [33]. In this research, the addition of RP-β-Ala to the diet notably boosted serum T-AOC levels, consistent with earlier studies indicating that β-Ala supplementation raised T-AOC in the plasma of broilers [10]. Furthermore, investigations have shown that a supplementation of 600 mg/kg β-Ala improved the activity of muscle glutathione peroxidase (GSH-Px) in Ningxiang pigs [11]. These results collectively indicate that β-Ala can markedly bolster the body’s antioxidant capacity. This enhancement may be attributed to β-Ala serving as a precursor for carnosine, a dipeptide known for its antioxidant properties [34,35]. TNF-α, a pro-inflammatory cytokine produced by mononuclear phagocytes, initiates an acute-phase protein response and can induce fever and inflammation when abnormally synthesized within tissues [36]. Our study observed that RP-β-Ala supplementation decreased TNF-α levels, correlating with the improved T-AOC values. Therefore, RP-β-Ala supplementation not only enhances the host’s antioxidant capacity but also mitigates inflammatory responses, highlighting its potential as a nutritional strategy to improve overall health and performance in livestock.

### 4.3. Effect of Supplementary Rumen-Protected β-Ala on Rumen Fermentation and Bacteria in Beef Cattle

Rumen fermentation characteristics (TVFA, NH_3_-N, MCP, and pH value) are crucial indicators of an animal’s physiological status. TVFA, the end product of rumen microbial fermentation, supplies approximately 70% of the energy requirements for ruminants [37]. Studies have shown that direct alanine supplementation does not affect rumen TVFA levels but significantly increases the abundance of *Prevotella* in the rumen [17]. However, in our study, dietary supplementation with RP-β-Ala significantly elevated the ruminal concentrations of TVFA, acetate, and propionate. This suggests that, unlike uncoated alanine, RP-β-Ala can directly influence VFA synthesis within the rumen. This discrepancy may be attributed to the rumen protection technology, which allows β-alanine to exert its effects after entering systemic metabolism via the hindgut. Specifically, β-alanine functions as a substrate for hepatic gluconeogenesis, thereby conserving propionate for glucose synthesis. Moreover, the terminal metabolite of β-alanine is acetate. Together, these mechanisms reduce the flux of both acetate and propionate from the rumen into the bloodstream. Consequently, this reduction in systemic absorption leads to increased concentrations of acetate, propionate, and TVFA within the rumen. Dietary proteins are degraded by microorganisms into peptides, amino acids, and NH_3_, with rumen NH_3_-N serving as the primary substrate for MCP synthesis. Hu et al. [17] found that alanine addition had no effect on NH_3_-N content but significantly increased MCP content, likely due to the direct utilization of nitrogen from alanine for MCP synthesis. In our study, RP-β-Ala supplementation increased rumen NH_3_-N and MCP levels. This outcome may be associated with an increased abundance of proteolytic bacteria, such as *Prevotella*, which facilitate the proteolysis of feed proteins and the generation of NH_3_. Additionally, ruminal NH_3_ partially originates from BUN that is recycled back into the rumen [1]. The observed reduction in serum BUN levels in this study may be correlated with a greater proportion of BUN being rerouted back to the rumen for MCP synthesis. Furthermore, BUN serves as an indicator of protein metabolism, reflecting amino acid balance and negatively correlating with protein utilization [38]. Even under rumen-protected conditions, β-Ala is indirectly recycled to the rumen through salivary carnosine → β-Ala + His → bacterial utilization and via reduced BUN/taurine flux; together, these subtle but sustained inputs shift microbial competition without altering total VFA or pH. This hypothesis is consistent with previous reports that RP amino acids modulate ruminal fermentation in cattle [5] and explains why RP-β-Ala still modulates ruminal parameters while remaining largely intact in situ. These results suggest that RP-β-Ala supplementation can improve rumen fermentation, thereby providing more energy for beef cattle growth.

The main contributors to the fermentation process in the rumen are bacteria. The indices of α-diversity indicate both the richness and variety of bacterial species present within the rumen. Our results demonstrated that dietary supplementation with RP-β-Ala significantly increased the Shannon index, indicating that it effectively enhanced rumen microbial diversity. This increase in diversity might result from reciprocal shifts in the relative abundances of specific bacterial phyla. Specifically, the relative abundance of the phylum *Firmicutes* decreased by approximately 9.99%, while the relative abundance of the phylum *Bacteroidetes* increased by approximately 11.04% in the RP-β-Ala group. The predominant core bacterial groups in the rumen of ruminants are *Firmicutes* and *Bacteroidetes*, characterized by their significant relative abundance. *Firmicutes* primarily consist of Gram-positive bacteria, notably featuring fiber-degrading genera like *Ruminococcus* and *Clostridium*, which are capable of breaking down fibers and cellulose [39]. Conversely, *Bacteroidetes*, notably *Prevotella*, excel in starch and non-cellulosic polysaccharide degradation, leveraging polysaccharide utilization loci to break down xylans, pectins, and proteins [40]. The present study shows that supplementation of RP-β-Ala significantly increased the relative abundance of the genus *Prevotella*. This bacterium is known for its ability to hydrolyze proteins and degrade hemicellulose, primarily producing acetate and propionate [41]. Additionally, *Prevotella* exhibits higher specific protease activity, which enhances the degradation of feed proteins in the rumen [42]. This increased proteolytic activity may be associated with the higher feed CP digestibility and the elevated contents of acetate and propionate observed in the study. In addition, RP-β-Ala supplementation also increased the genera *Treponema* and *Selenomonas*. In the rumen, a prevalent bacterial group known as *Treponema* is linked to the utilization of soluble carbohydrates [43], exhibiting a greater abundance in the rumen of beef cattle that are fed high-concentrate diets [44]. Numerous *Selenomonas* strains are capable of utilizing starch, primarily generating lactate, acetate, and propionate as their end products, and studies have indicated that these strains are more abundant in the rumen of cows with high feed efficiency compared to those with low feed efficiency [45]. The results from ruminal bacteria genera indicated that dietary supplementation with RP-β-Ala has a similar effect to increasing the digestion and utilization of nutrients. Moreover, similar to the influence at the genus level, *Selenomonas_ruminantium_subsp._lactilytica_TAM642*, *Prevotella_ruminicola*, *Prevotella_sp._BP1-145*, and *Selenomonas_ruminantium_AB3002*, which belong to the *Prevotella* and *Selenomonas* genera, were the top four species with upregulated levels in the RP-β-Ala group. On the other hand, RP-β-Ala supplementation decreased the levels of “harmful” bacteria such as *Mogibacterium*, *Thermoactinomyces vulgaris*, and *Saccharopolyspora_rectivirgula_g__norank*. Bacteria in the genus *Mogibacterium* are anaerobic, Gram-positive, non-fermentative bacteria. Previous studies have shown that *Mogibacterium* was significantly more abundant in the rumen of high-methane-producing cattle [46], and the increase in the abundance of *Mogibacterium* during high-grain feeding could have some deleterious effects on rumen epithelial health [47]. Exposure to *Saccharopolyspora rectivirgula*, previously referred to as *Micropolyspora faeni*, has the potential to lead to lung infections in both cattle and the farmers who work with them [48]. The organism’s presence triggers a microenvironment that promotes inflammation through the release of various cytokines (including IFN-γ, TNF-α, IL-1β, IL-6, IL-8, IL-10, IL-12, IL-13, and IL-17A) and chemokines, which attract inflammatory cells to the site [49].

### 4.4. Effect of Supplementary Rumen-Protected β-Ala on Rumen Metabolomics in Beef Cattle

Rumen metabolomics supports the interpretation of the rumen microbiota. Our data showed that the levels of spermine (C00750) and spermidine (C00315) were reduced in the RP-β-Ala group. Remarkably, a prior investigation revealed that β-Alanine effectively reduced biogenic amines in the muscle of broilers, aligning with the findings of our research [50]. Once introduced into the bloodstream, spermine and spermidine can undergo oxidation to form aldehydes and hydrogen peroxide via amine oxidase [51]. The generation of hydrogen peroxide from spermidine and spermine is attributed to the absence of catalase in the rumen, which is essential for hydrogen peroxide breakdown; an excessive buildup of this compound can disturb the homeostasis of the rumen microbiota [22]. In the current study, Spearman’s rank correlation analysis indicated a positive relationship between *Saccharopolyspora rectivirgula* and spermidine (Figure 6D). Additionally, another investigation noted that *Saccharopolyspora* was capable of synthesizing spermidine even when cultured in a synthetic medium devoid of polyamines [52]. Combining the above studies, it can be speculated that RP-β-Ala supplementation was associated with a lower relative abundance of *Saccharopolyspora* and a concomitant reduction in ruminal spermidine, but follow-up axenic cultures or stable-isotope labeling are required to establish causality. In lipid metabolism, it is worth paying attention to arachidonic acid (ARA) metabolism, including 11,12-DiHETrE (C14774), 14,15-DiHETrE (C14775), and Prostaglandin D2 (C00696). ARA and derivatives play a critical role in regulating inflammatory responses [53]. At present, there are three main types of enzymes involved in the metabolism of ARA, namely cyclooxygenase, lipoxygenase, and cytochrome P450. DiHETrE is derived from the hydrolysis of epoxyeicosatrienoic acids (EETs), which are primary metabolites of arachidonic acid generated via the cytochrome P450 epoxygenase pathway. A previous study indicated that 11,12-DiHETrE is a pro-inflammatory eicosanoid associated with increased pulmonary hypertension risk [54]. It was also the top candidate as a single biomarker for differentiating nonalcoholic fatty liver from nonalcoholic steatohepatitis [55], supporting its role in promoting inflammatory pathways. Conversely, 14,15-DiHETrE exhibits anti-inflammatory properties [56]. Another differential metabolite contained in ARA derivatives that exerts anti-inflammatory effects is prostaglandin D2. Prostaglandin D2 is biosynthesized from ARA via the cyclooxygenase pathway, where ARA is first converted to prostaglandin H2 by cyclooxygenase and subsequently isomerized to prostaglandin D2 by prostaglandin D synthase. By interacting with its receptors, prostaglandin D2 can inhibit the production of inflammatory cytokines and reduce the infiltration of inflammatory cells [57]. Our results revealed that 11,12-DiHETrE was downregulated, while 14,15-DiHETrE and prostaglandin D2 were upregulated in the RP-β-Ala group (Figure 5). Such regulation likely suppresses pro-inflammatory signaling and enhances anti-inflammatory immunity, thereby remodeling the inflammatory network and exerting comprehensive anti-inflammatory and protective effects. In addition, Spearman’s rank correlation analysis revealed that 14,15-DiHETrE exhibited positive correlations to *Prevotella_sp._BP1-145*, *Proteus_mirabilis*, and *Treponema_saccharophilum_DSM_2985*, and these three bacteria were negatively correlated with spermine, implying a significant role for these three bacteria in modulating inflammatory reaction.

## 5. Conclusions

This study demonstrates that dietary supplementation with RP-β-Ala promoted beef cattle growth by improving the body’s antioxidant capacity and reducing inflammation. This effect was partly related to changes in rumen microorganisms and metabolites. It enriches beneficial bacteria such as *Prevotella*, *Treponema*, and *Selenomonas* while inhibiting harmful bacteria such as *Thermoactinomyces*, *Saccharopolyspora*, and polyamine synthesis (spermine and spermidine). This optimizes rumen function, boosts host antioxidant capacity, reduces inflammation, and modulates key pathways like arachidonic acid metabolism (Figure 7).

## Figures and Tables

**Figure 1 animals-16-00043-f001:**
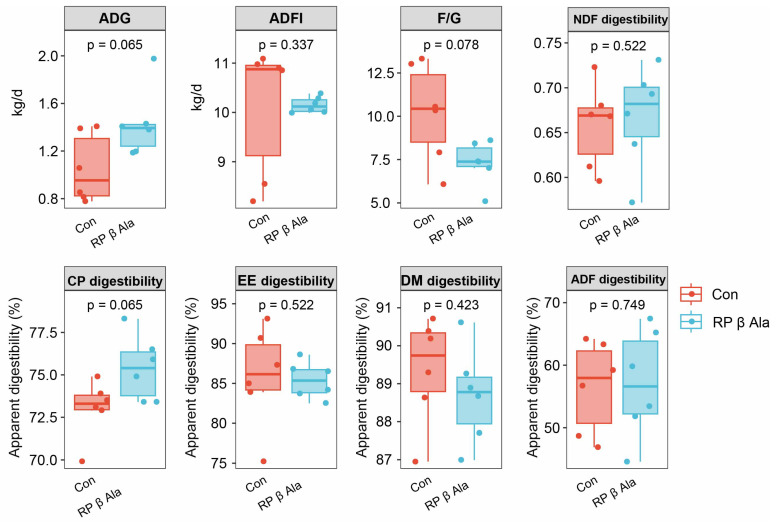
Effect of dietary supplementation with rumen-protected β-alanine (RP-β-Ala) on the growth performance and apparent nutrient digestibility of beef cattle. Con, the cattle fed with a basal diet; RP_beta_Ala, the cattle fed with a basal diet supplemented with RP-β-Ala (96 g/d/cattle). ADG = average daily gain; ADFI = average daily feed intake; F/G = ADFI/ADG; CP = crude protein; EE = ether extract; DM = dry matter; NDF = neutral detergent fiber; ADF = acid detergent fiber.

**Figure 2 animals-16-00043-f002:**
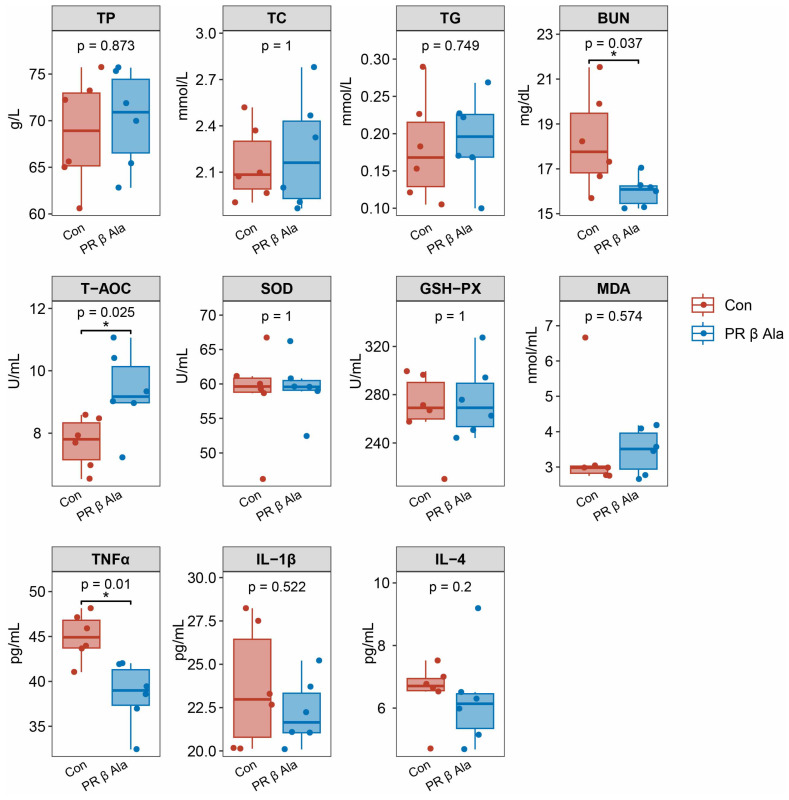
Effects of dietary supplementation with rumen-protected β-alanine (RP-β-Ala) on the serum parameters of beef cattle. Con, the cattle fed with a basal diet; RP-β-Ala, the cattle fed with a basal diet supplemented with RP-β-Ala (96 g/d/cattle). TP = total protein; TC = total cholesterol; TG = triglycerides; BUN = blood urea nitrogen; T-AOC = total antioxidant capacity; SOD = superoxide dismutase; GSH-Px = glutathione peroxidase; MDA = malondialdehyde; TNF-α = tumor necrosis factor-α; IL-1β = interleukin-1β; IL-4 = interleukin-4. * *p* < 0.05.

**Figure 3 animals-16-00043-f003:**
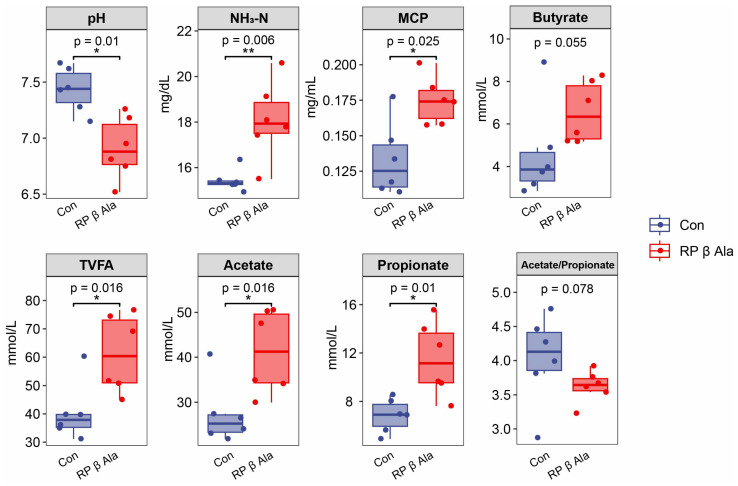
Effects of dietary supplementation with rumen-protected β-alanine (RP-β-Ala) on the rumen fermentation indicators of beef cattle. Con, the cattle fed with a basal diet; RP-β-Ala, the cattle fed with a basal diet supplemented with RP-β-Ala (96 g/d/cattle). NH_3_-N = ammonia nitrogen; MCP = microbial crude protein; VFA = volatile fatty acids. * *p* < 0.05, ** *p* < 0.01.

**Figure 4 animals-16-00043-f004:**
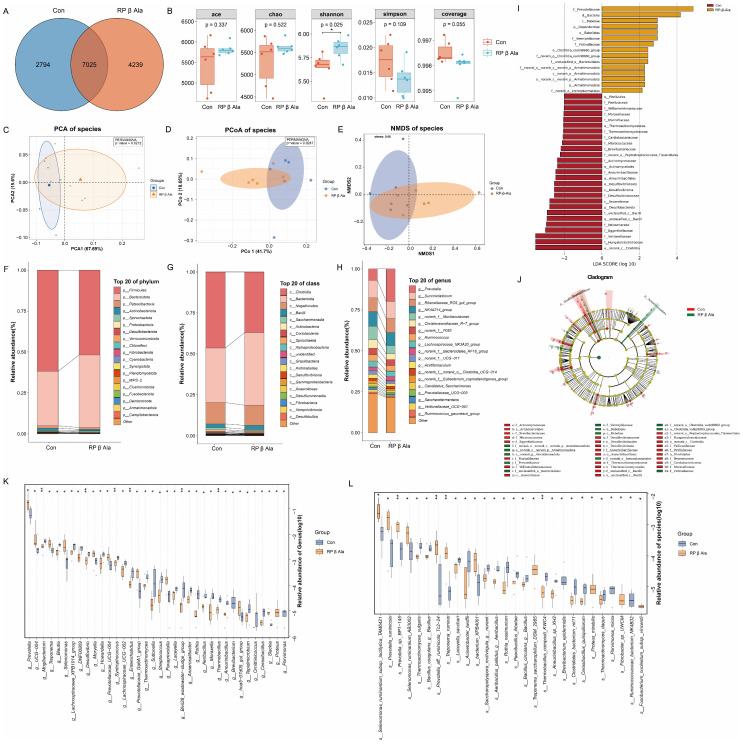
Effects of dietary supplementation with rumen-protected β-alanine (RP-β-Ala) on the ruminal microbial community. (**A**) Venn diagram of OTU statistics of rumen bacteria. (**B**) The α-diversity of rumen bacteria. (**C**) Principal component analysis (PCA) score plots. (**D**) Principal coordinate analysis (PCoA) score plots. (**E**) Non-metric multidimensional scaling (NMDS) score plots. (**F**) Phyla-, (**G**) classes-, and (**H**) genera-level composition of the rumen bacteria (top 20). (**I**) Histograms of linear discriminant analysis (LDA) for the differential biomarker. (**J**) LEfSe analysis indicated the biomarker bacteria. (**K**,**L**) At the genus and species level, there was a significant difference between groups. * *p* < 0.05, ** *p* < 0.01.

**Figure 5 animals-16-00043-f005:**
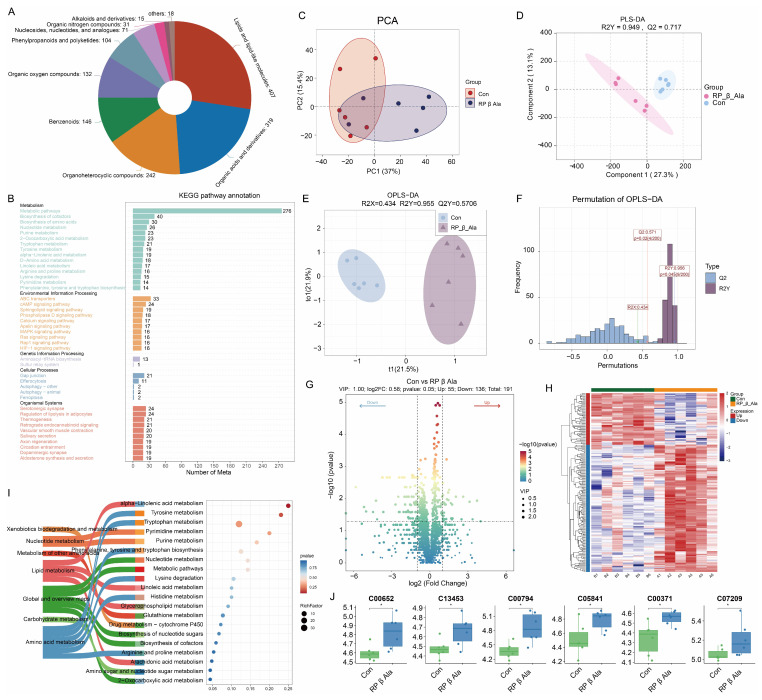
Effects of dietary supplementation with rumen-protected β-alanine (RP-β-Ala) on the ruminal metabolomic features in beef cattle. (**A**) Pie chart showing metabolite classification. (**B**) Bar chart of metabolite KEGG function annotation. (**C**) Principal component analysis (PCA) score plot. (**D**) Partial least squares discriminant analysis (PLS-DA) score plot. (**E**,**F**) Orthogonal partial least squares discriminant analysis (OPLS-DA) score plot and OPLS-DA permutation test. (**G**) Volcano plot of the annotated metabolites. (**H**) Heatmap of differential accumulated metabolites. (**I**) KEGG enrichment pathways based on the differential metabolites; the *X*-axis represents the enrichment factor, which represents the number of differential metabolites in each pathway divided by the number of all metabolites in that pathway, and the *Y*-axis represents the names of pathways that are significantly enriched. (**J**) Violin plot analysis of 6 differential metabolites between control and RP-β-Ala groups. * *p* < 0.05.

**Figure 6 animals-16-00043-f006:**
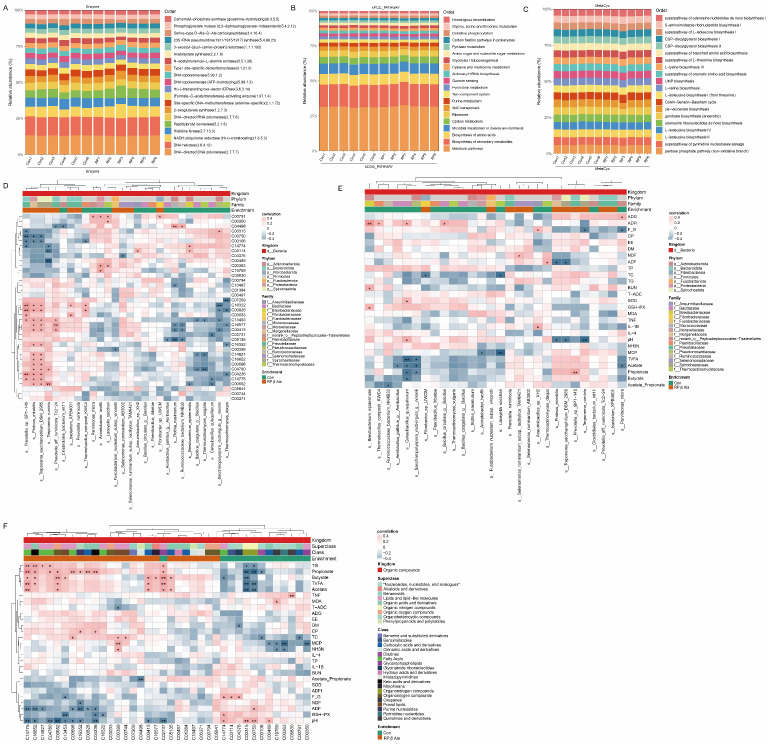
Predicted bacterial community functions and Spearman’s correlation analysis. Functional pathway abundance prediction of bacteria of each sample in the rumen against enzyme (**A**), KEGG categories (**B**), and MetaCyc (**C**) by PICRUSt2. (**D**) Correlation analysis between differential bacterial species and differential metabolites. (**E**) Correlation analysis between differential bacterial species and phenotypic data. (**F**) Correlation analysis between differential metabolites and phenotypic data. * *p* < 0.05, ** *p* < 0.01.

**Figure 7 animals-16-00043-f007:**
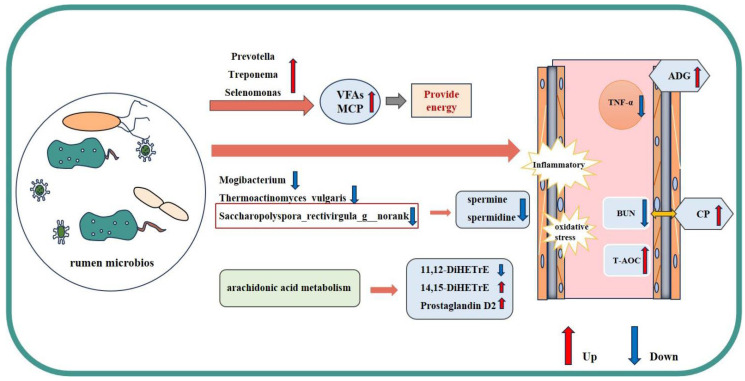
Overview of the full text. The rumen microbial community and metabolome of the two groups of beef cattle were analyzed. Dietary supplementation with RP-β-Ala increased the abundance of beneficial bacteria, such as *Prevotella*, *Treponema*, and *Selenomonas*, which improved ruminal fermentation, optimized rumen fermentation capacity, and provided more energy for beef cattle growth. The abundance of harmful bacteria, such as *Thermobifida*, *Thermoactinomyces vulgaris*, and *Saccharopolyspora,* was inhibited. Moreover, a significant positive correlation was observed between *Saccharopolyspora* and spermidine, indicating that this bacterial genus may be an important contributor to the ruminal polyamine pool. Additionally, the arachidonic acid metabolism pathway was modulated. This suggests that RP-β-Ala could stimulate beef cattle growth by boosting antioxidant capacity and reducing inflammation.

**Table 1 animals-16-00043-t001:** Composition and nutrient levels of basal diet (air-dry basis, %).

Ingredients	Content	Nutrients	Content
Brewer’s grains	29.74	Dry matter	63.97
Rice straw silage	25.95	Crude protein	14.35
Corn	21.84	Crude fat	4.68
Soybean meal	6.38	Ash	8.39
Whet Straw	5.74	Neutral detergent fiber	32.65
Wheat bran	2.95	Acid detergent fiber	13.23
Microbial agent	4.73		
NaHCO_3_	0.7		
Premix ^1^	1.97		
Total	100		

Note: ^1^ The premix provided per kg of diet was as follows: 80,000 IU of vitamin A, 20,000 IU of vitamin D3, 320 IU of vitamin E, 3100 mg of Fe, 1500 mg of Mn, 2000 mg of Zn, 650 mg of Cu, 15 mg of Co, 40 mg of I, 115 g of Ca and 35 g of P.

## Data Availability

The data are available from the corresponding author on reasonable request. Sequence data in this study were uploaded to the NCBI SRA database (PRJNA1208407).

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
