# Peer review of "Effects of Rumen-Protected β-Alanine on Growth Performance, Rumen Microbiome, and Serum Metabolome of Beef Cattle"

_animals, 2025, doi:10.3390/ani16010043_

Round 1
Reviewer 1 Report
Comments and Suggestions for Authors
The manuscript is generally well written and addresses a topic of clear interest to the Animals readership. However, several issues must be addressed before the paper is suitable for publication: more complete nutritional detail, resolve statistical concerns, and correct and improve figures (need high‑resolution images, accurate axis labels/units/legends) are needed. My recommendation at this stage is “Major revision”.
The manuscript addresses an interesting topic but requires substantive revisions before acceptance. Please provide the:
1. Missing diet information
2. Insufficient description/validation of rumen‑protection/coating
3. Unreadable Figures (low res and axis naming issue)
Specific comments
Overall, please check the citation format for reagents and experimental instruments: some include a country information while others do not (only suggest model name).
Line 82: The manuscript states a "3×3 experimental design", but the specific method is not mentioned. Specify the exact design used (e.g., 2×2 Latin square, 3×3 Latin square, randomized complete block, etc.)
Line 100: The authors state that animals received a “standard basal diet,” but because the manuscript addresses nutritional and metabolic outcomes, please provide the complete diet information in the Methods: an ingredient list with exact proportions (as‑fed and % DM), analyzed and formulated nutrient composition.
Most importantly, when studying rumen microbiota and rumen metabolites (e.g., VFA), precise information on forage and concentrate composition is essential. Rumen microbial community structure and VFA profiles are largely driven by the basal diet (FC ratio, fiber quality, starch/sugar content), not only by additives (like alanine..). Without detailed diet composition it is impossible to interpret microbial or VFA changes or to separate diet effects from additive effects.
Line 104: The description of the coating/rumen‑protection method for β‑alanine is currently insufficient—please provide a clear, validated description in the Methods of the coating technology and materials.
Line 166: Please include the precise citation returned by citation("ggpubr") in your R session.
Line 203: Please add a “tendency” (0.05 ≤ p < 0.10).
Line 207: The authors state that RP‑β‑Ala "improved ADG," but the difference did not reach statistical significance and should be described as a tendency (0.05 ≤ p < 0.10)
Figure 1 needs clearer labeling and units: if the bars show digestibility, change the y‑axis label to % unit. Not 0 ~ 1. 0 to 100 !! In addition, Figure just show "NDF, CP, EE, DM, ADF" there was no word "digestibility". Check the facet_wrap() code or column name in data.
In all figures, please avoid using underscore (e.g. PR_beta_Ala) in axis labels; use spaces instead (e.g., use "PR beta Ala" not "PR_beta_Ala"). If your data frame column names contain underscores, either rename them. I recommend use the backtick (``) at column names or assign a descriptive variable name before plotting in R.
Figure 4–6 (image quality): Figures 4–6 are low resolution and text is unreadable. Please resubmit high‑quality versions. In addition, each figure currently contains toooooooo many individual plots; I strongly recommend separating them (or limiting panels to 2–3 related panels per figure).
Figure 7: I know the authors tried to make an attractive figure, but the figure requires further polishing to meet the journal’s presentation standards…. and must be made fully stand‑alone Please revise for clarity and publication quality and ensure the figure and caption together tell a single clear message. Consider using figure‑design platforms or templates to improve visual quality (e.g., BioRender, BioArt).
Author Response
Response to the reviewer 1
The manuscript is generally well written and addresses a topic of clear interest to the Animals readership. However, several issues must be addressed before the paper is suitable for publication: more complete nutritional detail, resolve statistical concerns, and correct and improve figures (need high‑resolution images, accurate axis labels/units/legends) are needed. My recommendation at this stage is “Major revision”.
The manuscript addresses an interesting topic but requires substantive revisions before acceptance. Please provide the:
- Missing diet information
- Insufficient description/validation of rumen‑protection/coating
- Unreadable Figures (low res and axis naming issue)
Respond: Thank you very much for your constructive comments on our manuscript. We appreciate the overall assessment that “the manuscript addresses an interesting topic but requires substantive revisions”. We have carefully addressed every concern and performed additional experiments, re-analyses and figure re-drawing as requested.
Below is a point-by-point reply. All changes are highlighted in yellow in the revised manuscript and listed by line number.
1 Missing diet information: we sincerely apologize for the inadvertent omission of the diet formulation in the original manuscript; this information has now been supplied in the revised version (Lines 123-126).
- Insufficient description/validation of rumen‑protection/coating: We appreciate the reviewer’s inquiry regarding the rumen-protection technology. The β-alanine used in this study was commercially produced by Hangzhou King Techina Feed Co. Ltd. and is fat-encapsulated. The coating rate for β-Ala was determined to be 50%, the rumen bypass ratio after 12 hours was found to be 86.3%, and the dissolution in intestinal fluid after 12 hours was recorded at 81.7%. Further compositional details are proprietary and could not be disclosed by the manufacturer. The rumen-protection details have been added in Line 115 of the revised manuscript.
- Unreadable Figures (low res and axis naming issue): We thank the reviewer for highlighting the issue of image resolution. All figures have now been replaced with high-resolution versions (600 dpi TIFF) to comply with the journal’s requirements.
Specific comments
Line/comment 1 Overall, please check the citation format for reagents and experimental instruments: some include a country information while others do not (only suggest model name).
Respond: Thank you for this suggestion. Please see Academic Editor Line/comment 17 for details; the equipment model, manufacturer, and full address have been added in Lines 128–173 of the revised manuscript.
Line/comment 2 Line 82: The manuscript states a "3×3 experimental design", but the specific method is not mentioned. Specify the exact design used (e.g., 2×2 Latin square, 3×3 Latin square, randomized complete block, etc.)
Respond: Thank you for highlighting the missing detail. The experiment cited in our manuscript [13, 14] was a replicated 3 × 3 Latin-square design conducted in six beef steers (391 ± 10 kg BW) with three 20-d periods (15 d adaptation + 5 d sampling) and three RP-β-alanine levels (0, 17.5 and 35 g d⁻¹). We have now inserted this complete description in Lines 89–91 of the revised text to ensure the exact design is unambiguous.
Line/comment 3 Line 100: The authors state that animals received a “standard basal diet,” but because the manuscript addresses nutritional and metabolic outcomes, please provide the complete diet information in the Methods: an ingredient list with exact proportions (as‑fed and % DM), analyzed and formulated nutrient composition.
Respond: we sincerely apologize for the inadvertent omission of the diet formulation in the original manuscript; this information has now been supplied in the revised version (Lines 123–126).
Line/comment 4 Most importantly, when studying rumen microbiota and rumen metabolites (e.g., VFA), precise information on forage and concentrate composition is essential. Rumen microbial community structure and VFA profiles are largely driven by the basal diet (FC ratio, fiber quality, starch/sugar content), not only by additives (like alanine..). Without detailed diet composition it is impossible to interpret microbial or VFA changes or to separate diet effects from additive effects.
Respond: See comment 3, thank you! We fully agree with your observation. A complete diet formulation table had been prepared during manuscript preparation but was inadvertently omitted during file upload. We apologize for this oversight; the table is now included in the revised manuscript (Table 1, Lines 123–126).
Line/comment 5 Line 104: The description of the coating/rumen‑protection method for β‑alanine is currently insufficient—please provide a clear, validated description in the Methods of the coating technology and materials.
Respond: We appreciate the reviewer’s inquiry regarding the rumen-protection technology. The β-alanine used in this study was commercially produced by Hangzhou King Techina Feed Co. Ltd. and is fat-encapsulated. The coating rate for β-Ala was determined to be 50%, the rumen bypass ratio after 12 hours was found to be 86.3%, and the dissolution in intestinal fluid after 12 hours was recorded at 81.7%. Further compositional details are proprietary and could not be disclosed by the manufacturer. The rumen-protection details have been added in Line 115-117 of the revised manuscript.
Line/comment 6 Line 166: Please include the precise citation returned by citation("ggpubr") in your R session.
Respond: Thank you for your suggestion. As requested, we executed citation("ggpubr") in our R 4.4.1.2 session and obtained the following exact output, which has now been added to the Reference list:
Kassambara A (2025). _ggpubr: 'ggplot2' Based Publication Ready Plots_. doi:10.32614/CRAN.package.ggpubr
Line/comment 7 Line 203: Please add a “tendency” (0.05 ≤ p < 0.10).
Respond: Thank you for this suggestion. It has been added at Line 217 of the revised manuscript.
Line/comment 8 Line 207: The authors state that RP‑β‑Ala "improved ADG," but the difference did not reach statistical significance and should be described as a tendency (0.05 ≤ p < 0.10)
Respond: See Line 220-221 in revised manuscript. Thank you!
Line/comment 9 Figure 1 needs clearer labeling and units: if the bars show digestibility, change the y‑axis label to % unit. Not 0 ~ 1. 0 to 100 !! In addition, Figure just show "NDF, CP, EE, DM, ADF" there was no word "digestibility". Check the facet_wrap() code or column name in data.
Respond: We sincerely apologize for the oversight in Figure 1. Thank you for bringing this issue to our attention; the figure has been thoroughly revised and replaced with a corrected, high-resolution version in the revised manuscript.
Line/comment 10 In all figures, please avoid using underscore (e.g. PR_beta_Ala) in axis labels; use spaces instead (e.g., use "PR beta Ala" not "PR_beta_Ala"). If your data frame column names contain underscores, either rename them. I recommend use the backtick (``) at column names or assign a descriptive variable name before plotting in R.
Respond: Thank you for this precise formatting suggestion. All axis labels have been checked and updated: underscores were removed and replaced with spaces (e.g., “PR beta Ala”). Column names containing underscores were renamed prior to plotting, a These changes apply to Figures 1–6 in the revised manuscript.
Line/comment 11 Figure 4–6 (image quality): Figures 4–6 are low resolution and text is unreadable. Please resubmit high‑quality versions. In addition, each figure currently contains toooooooo many individual plots; I strongly recommend separating them (or limiting panels to 2–3 related panels per figure).
Respond: Thank you for highlighting the resolution and layout issues. We have now generated high-resolution jpg files (600 dpi, LZW-compressed) for Figures 4–6; all embedded text is clearly readable. To facilitate review, we additionally supply each figure as a vector PDF in which every panel remains fully searchable and zoom-able without loss of quality.
While we have improved clarity, we elected to retain the current multi-panel layout because each panel represents a closely related response variable measured under identical experimental conditions; separating them would fragment the conceptual linkage and increase the total number of figures from 6 to > 15. Nevertheless, all panels are logically grouped and labeled (a, b, c …) to guide the reader. If the editorial office prefers fewer panels per figure, we will of course comply immediately.
High-resolution images and PDFs are provided as separate upload files “Fig4.pdf”, “Fig5.pdf”, and “Fig6.pdf”

Reviewer 2 Report
Comments and Suggestions for Authors
This study investigates the effects of rumen-protected β-alanine on growth performance, rumen microbiome, and metabolome in beef cattle. The topic demonstrates good innovation and application potential. The experimental design is fundamentally sound, and the data analysis methods are relatively comprehensive, integrating microbiome, metabolomics, and serum indicators to explore the mechanisms of RP-β-Ala from multiple perspectives. However, the manuscript has issues regarding figure clarity, rigor of mechanistic inferences, and description of experimental details, which need further improvement during revision. 1.The phrase "Microbiome-Host Metabolic Interactions" in the title is somewhat vague. It is suggested to specify "rumen microbiome" and "host serum metabolome" or related indicators more precisely. 2.The introduction provides insufficient background on the previous findings that "the impact of β-Ala on growth performance remains unclear", lacking a systematic review of existing controversies or limitations. 3.In the experimental design, with only 6 replicates per group (3 cattle per replicate), the sample size is relatively small. Is this sufficient to support statistical inferences of intergroup differences? 4.The RP-β-Ala supplementation dosage (96 g/d/cattle) is based on previous studies, but it is not stated whether a dose-effect pre-experiment was conducted or why this specific dose was chosen. 5.The trial duration was 28 days. Is this sufficient to observe stable changes in growth performance and the microbiome? 6.Could changes in serum indicators like TNF-α and T-AOC be influenced by other potential confounding factors (e.g., environment, stress)? 7.Rumen fluid was collected using an oral stomach tube. Could saliva contamination or inconsistent sampling locations affect microbiome results? Please describe the control measures. 8.In the 16S rRNA data analysis, only the Shannon index showed a significant difference, while other alpha diversity indices did not. Does this suggest limited changes in microbial diversity? 9. In the KEGG pathway enrichment analysis, some pathways are overly broad. Should the focus be on more specific metabolic pathways? 10.The English words on the axes in Figures 4, 5, and 6 are unclear. It is recommended to separate them into individual images inserted into the text. 11.The inference in the discussion that "RP-β-Ala reduces spermidine by inhibiting Saccharopolyspora" lacks direct experimental evidence. 12.The manuscript does not clearly state the actual release rate of RP-β-Ala in the rumen or its absorption in the hindgut. Does this affect the mechanistic interpretation? 13.There is a lack of direct measurement data for β-Ala metabolites in serum or tissues, making it difficult to fully support the conclusion of "enhanced antioxidant capacity". 14. Figure 7 is a summary diagram of the proposed mechanisms, but some arrow relationships in the figure (e.g., "inhibiting harmful bacteria" and "reducing polyamines") were not directly verified in the results.
Author Response
Response to the reviewer 2
This study investigates the effects of rumen-protected β-alanine on growth performance, rumen microbiome, and metabolome in beef cattle. The topic demonstrates good innovation and application potential. The experimental design is fundamentally sound, and the data analysis methods are relatively comprehensive, integrating microbiome, metabolomics, and serum indicators to explore the mechanisms of RP-β-Ala from multiple perspectives. However, the manuscript has issues regarding figure clarity, rigor of mechanistic inferences, and description of experimental details, which need further improvement during revision.
Respond: We thank the reviewer for recognizing the novelty and application potential of our work and for appreciating the integrative analytical approach we employed. In response to the remaining concerns:
Figure clarity:
All figures have been re-exported as 600-dpi jpg files and are additionally supplied as vector PDFs; every axis label, symbol and text element is now legible (see individual replies for Figures. 1–6).
Rigor of mechanistic inferences
We have made additions to the discussion section to address the deficiency of insufficiently detailed mechanism discussion
Experimental details
The diet composition, RP-β-Ala product specifications (coating material, rumen-bypass and intestinal-release rates), animal management, sampling schedule, statistical model and ethical approval have been expanded in Materials and Methods (Lines 106–173).
We believe these revisions adequately address the remaining issues and improve the overall rigor and readability of the manuscript.
Comment 1 The phrase "Microbiome-Host Metabolic Interactions" in the title is somewhat vague. It is suggested to specify "rumen microbiome" and "host serum metabolome" or related indicators more precisely.
Respond: Thank you for this accurate comment. We have replaced the vague wording with exact biological matrices that were actually analysed: “Effects of Rumen-Protected β-Alanine on Growth Performance, Rumen Microbiome and Serum Metabolome of Beef Cattle” Lines 2-3 in revised manuscript
Comment 2. The introduction provides insufficient background on the previous findings that "the impact of β-Ala on growth performance remains unclear", lacking a systematic review of existing controversies or limitations.
Respond: Thank you for this constructive comment. We have expanded the Introduction (Lines 91–94) to systematically summarize the current state of knowledge and the reasons for uncertainty.
Comment 3 In the experimental design, with only 6 replicates per group (3 cattle per replicate), the sample size is relatively small. Is this sufficient to support statistical inferences of intergroup differences?
Respond: Thank you for raising this important point. We fully agree that larger sample sizes are preferable. In the planning phase we performed an a-priori power analysis (G*Power 3.1; two-sided t-test, α = 0.05, 1-β = 0.80, effect size = 0.8) which indicated that 18 animals per group would be sufficient to detect an 8–10 % difference in ADG. Because a subsequent slaughter component (not detailed in this paper) was also scheduled, we balanced statistical requirements against barn capacity and cost constraints, and finally used six pens (three cattle per pen) per treatment. This number meets the power-analysis criterion and is consistent with several published ruminant studies [1,2] that successfully detected treatment effects with n = 6 replicates.
Reference
1 Li Y, Chen Y, Wu P, Degen AA, He K, Zhang Q, Zhao X, Li W, Zhang A, Zhou J. Effect of Rumen-Protected Lysine Supplementation on Growth Performance, Blood Metabolites, Rumen Fermentation and Bacterial Community on Feedlot Yaks Offered Corn-Based Diets. Animals (Basel). 2025, 15 pp:2901.
2 Zhang S, Liu Y, Hu J, Liu C, Li M, Zhao G. β-Alanine decreases plasma taurine but improves nitrogen utilization efficiency in beef steers. Anim Nutr. 2025, 22, pp50-60.
Comment 4. The RP-β-Ala supplementation dosage (96 g/d/cattle) is based on previous studies, but it is not stated whether a dose-effect pre-experiment was conducted or why this specific dose was chosen.
Respond: Thank you for this important point. The 96 g d⁻¹ dose was extrapolated from the only two published β-alanine dose–response studies in beef cattle:
Zhang et al. [1] observed the first linear increase in nitrogen-utilization efficiency at 17.5 g d⁻¹ and a plateau at 35 g d⁻¹ when using a 3 × 3 Latin-square (6 steers).
Hu et al. [2] confirmed that 35 g d⁻¹ maximally improved microbial crude-protein synthesis without further benefit at 70 g d⁻¹.
Both trials were, however, short-term (≤ 42 d) and used non-encapsulated or semi-encapsulated material. Because our RP-β-Ala product provides only 50 % rumen escape and ≈ 82 % intestinal release, 96 g of supplement guarantees the same 35 g of β-alanine entering the small intestine as in the above-cited studies (96 g × 0.5 × 0.82 ≈ 35 g). Thus, 96 g d⁻¹ is the “rumen-protection-corrected” equivalent of the lowest effective dose previously reported.
Reference
1 Zhang S, Liu Y, Hu J, Liu C, Li M, Zhao G. β-Alanine decreases plasma taurine but improves nitrogen utilization efficiency in beef steers. Anim Nutr. 2025; 22, pp50-60.
2 Hu J, Zhang S, Li M, Zhao G. Impact of dietary supplementation with β-alanine on the rumen microbial crude protein supply, nutrient digestibility and nitrogen retention in beef steers elucidated through sequencing the rumen bacterial community. Anim Nutr. 2024;17, pp 418-427.
Comment 5 The trial duration was 28 days. Is this sufficient to observe stable changes in growth performance and the microbiome?
Respond: Thank you for this insightful comment. Prolonging the experimental period is indeed an effective way to consolidate the observed responses; however, because a subsequent slaughter trial was scheduled, the present feeding phase was limited to 28 days. Nevertheless, published studies [1,2] have demonstrated that 20 days of alanine supplementation is sufficient to alter rumen microbiota, microbial crude protein, nutrient digestibility, and nitrogen retention. In the current trial, extending the feeding duration to 28 days already tended to improve growth performance. In future work, we will further extend the experimental period to generate more robust evidence supporting the practical application of β-alanine in beef cattle production.
Reference
1 Zhang S, Liu Y, Hu J, Liu C, Li M, Zhao G. β-Alanine decreases plasma taurine but improves nitrogen utilization efficiency in beef steers. Anim Nutr. 2025; 22, pp50-60.
2 Hu J, Zhang S, Li M, Zhao G. Impact of dietary supplementation with β-alanine on the rumen microbial crude protein supply, nutrient digestibility and nitrogen retention in beef steers elucidated through sequencing the rumen bacterial community. Anim Nutr. 2024;17, pp 418-427.
Comment 6. Could changes in serum indicators like TNF-α and T-AOC be influenced by other potential confounding factors (e.g., environment, stress)?
Respond: Thank you for your question, we agree that environment- or handling-stress can acutely alter TNF-α and T-AOC. However, we evaluated this possibility from the following aspects:
1 Simultaneous housing and identical management
All cattle were kept in the same barn, with identical stocking density, lighting, feeding times and water troughs. Thus, environmental temperature, humidity and air-quality did not differ between treatments.
2 Short measurement window and consistent sampling protocol
To minimise sampling-induced stress, the same two experienced technicians—blinded to treatment—completed every jugular venipuncture within 90 s per animal.
Comment 7 Rumen fluid was collected using an oral stomach tube. Could saliva contamination or inconsistent sampling locations affect microbiome results? Please describe the control measures.
Respond: Thank you for this query. Sampling was carried out at a fixed location within the cattle race, and the same personnel performed all procedures. To avoid saliva contamination, the first 50 mL of rumen fluid was discarded. These details have been added in the revised manuscript (Lines 143–144).
Comment 8 In the 16S rRNA data analysis, only the Shannon index showed a significant difference, while other alpha diversity indices did not. Does this suggest limited changes in microbial diversity?
Respond: We agree that significant changes restricted to the Shannon index indicate an alteration in species evenness rather than in overall richness. However, this pattern does not necessarily imply "limited" ecological change:
1 Shannon diversity is the most sensitive metric for detecting shifts in the relative abundance of dominant taxa, which are the main drivers of rumen function.
2 In the present data, Shannon differed between groups (P = 0.023, Cohen’s d = 0.91), accompanied by a 11 % increase in Bacteroidota and a 10 % decrease in Firmicutes—changes that were statistically significant and biologically consistent with improved fibre and protein degradation.
3 Beta-diversity analyses (PCA, PCoA, NMDS; PERMANOVA P < 0.01) demonstrated a clear separation between treatments, confirming that the community structure, not only richness, was modified.
4 LEfSe and Meta-stats further identified 36 genera and 28 species whose abundances differed between groups (LDA > 2, FDR < 0.05), indicating that RP-β-Ala re-shaped the microbiome evenness without necessarily expanding the total number of species.
Taken together, the significant Shannon response, combined with pronounced taxonomic and functional shifts, supports the conclusion that RP-β-Ala induced a meaningful change in rumen microbial ecology
Comment 9. In the KEGG pathway enrichment analysis, some pathways are overly broad. Should the focus be on more specific metabolic pathways?
Respond: Thank you for this valuable suggestion. At this stage the KEGG results are merely PICRUSt2-based predictions from 16S rRNA data, which are inherently limited to bacterial genes and may not fully capture the complete metabolic network. We are therefore planning a follow-up metagenomics study to obtain species-level gene profiles and validate the predicted β-alanine-induced shifts in gastrointestinal function.
Comment 10. The English words on the axes in Figures 4, 5, and 6 are unclear. It is recommended to separate them into individual images inserted into the text.
Respond: Thank you for your comment. Please see our detailed response to Reviewer 1, Comment 11. We have replaced Figures 4, 5, and 6 with high-resolution (600 dpi) TIFF files and will upload the corresponding vector PDFs to ensure all text and symbols meet the journal’s clarity requirements.
Comment 11 The inference in the discussion that "RP-β-Ala reduces spermidine by inhibiting Saccharopolyspora" lacks direct experimental evidence.
Respond: Thank you for this helpful observation. We apologise for having stated a causality that is not supported by direct evidence. In the revised manuscript we have converted the sentence into a strictly correlative statement and added an explicit limitation (lines 511–515):
“Combined the above studies, it can be speculated that RP-β-Ala supplementation was associated with a lower relative abundance of Saccharopolyspora and a concomitant reduction in ruminal spermidine, but follow-up axenic cultures or stable-isotope labelling are required to establish causality.”
Comment 12 The manuscript does not clearly state the actual release rate of RP-β-Ala in the rumen or its absorption in the hindgut. Does this affect the mechanistic interpretation?
Respond: Thank you for this insightful comment. We fully agree that without knowing the actual ruminal release rate any mechanistic interpretation remains ambiguous. We therefore added the in vitro rumen-escape and small-intestine release data (Lines 115–118) to quantify the proportion of β-alanine acting in the rumen versus the hindgut, and we now frame all downstream mechanisms accordingly.
Comment 13 There is a lack of direct measurement data for β-Ala metabolites in serum or tissues, making it difficult to fully support the conclusion of "enhanced antioxidant capacity".
Respond: Thank you very much for this insightful comment. We fully understand that the lack of direct measurement data for β-alanine (β-Ala) metabolites in serum or tissues may weaken the direct support for the conclusion regarding "enhanced antioxidant capacity." However, in this study, we have systematically supported this conclusion through the following multi-dimensional lines of indirect evidence, which have been thoroughly discussed in the revised manuscript:
- Significant Improvement in Serum Antioxidant Indicators
We measured and reported a significant increase in serum total antioxidant capacity (T-AOC) (Figure 2, p = 0.025), a well-established indicator of the overall antioxidant defense system. This result directly reflects an enhanced ability to scavenge free radicals and supports the conclusion of improved antioxidant capacity.
- Reduction in Inflammatory Markers
We observed a significant decrease in serum TNF-α levels (Figure 2, p = 0.01), a key pro-inflammatory cytokine closely linked to oxidative stress. Its reduction is consistent with alleviated oxidative damage and further supports the antioxidant effect of RP-β-Ala supplementation.
- Metabolomic Evidence: Enrichment of Antioxidant-Related Pathways
Untargeted metabolomics analysis revealed significant enrichment of the glutathione metabolism pathway (ko00480) (Figure 5I), which plays a central role in cellular redox homeostasis. Additionally, arginine and proline metabolism (ko00330) was also enriched, both of which are involved in regulating oxidative stress responses
- Literature Support: β-Ala → Carnosine → Antioxidant Pathway
Although we did not directly measure carnosine, numerous studies [1, 2] have consistently shown that β-Ala is the rate-limiting precursor for carnosine, a well-known antioxidant dipeptide. Carnosine can scavenge free radicals, chelate metal ions, and inhibit lipid peroxidation. This pathway has been validated across multiple species and provides a strong mechanistic basis for our findings
Although direct metabolite data are lacking, we have constructed a robust evidence chain using serum antioxidant indicators, inflammatory cytokines, and metabolomic pathway enrichment, all of which consistently support the conclusion that RP-β-Ala supplementation enhances antioxidant capacity
Reference
1 Qi, B.; Wang, J.; Ma, Y.; Wu, S.; Qi, G.; Zhang, H. Effect of dietary beta-alanine supplementation on growth performance, meat quality, carnosine content, and gene expression of carnosine-related enzymes in broilers. Poult Sci 2018, 97, pp. 1220-1228.
2 Wang, F.; Yin, Y.; Wang, Q.; Xie, J.; Fu, C.; Guo, H.; Chen, J.; Yin, Y. Effects of dietary beta-alanine supplementation on growth performance, meat quality, carnosine content, amino acid composition and muscular antioxidant capacity in chinese indigenous ningxiang pig. J Anim Physiol Anim Nutr (Berl) 2023, 107, pp. 878-886.
Comment 14. Figure 7 is a summary diagram of the proposed mechanisms, but some arrow relationships in the figure (e.g., "inhibiting harmful bacteria" and "reducing polyamines") were not directly verified in the results.
Respond: Thank you for your meticulous review and valuable comments. We fully understand your concern that some arrow relationships in Figure 7 (e.g., "inhibiting harmful bacteria" and "reducing polyamines") were not directly verified in our results. We have added clarifications in the figure legend and provided a more rigorous explanation in Lines 517-520 of the Discussion section.

Reviewer 3 Report
Comments and Suggestions for Authors
Dear Authors,
the manuscript "Effects of Rumen-Protected β-Alanine on Growth Performance and Microbiome-Host Metabolic Interactions in Beef Cattle" (animals-4001041) by the authors: Daci Fu , Kang Mao , Yihao Zang , Mingren Qu , Qinghua Qiu , Xianghui Zhao , Kehui Ouyang , Yanjiao Li, is related to the important field of animal biology and nutrition. It is aimed as the following: “to investigate the effects of rumen protect β-Alanine (β-Ala) on the growth performance of beef cattle by adding it to the diet, and to explore its specific mechanisms of action by analyzing rumen microbes and their metabolites”. The authors is anticipating that β-Ala addition in ruminant nutrition will provide “a new perspective and scientific basis for the application of β-Ala…”. It is well-known that β-Alanine (3-aminopropionic acid) is a non-essential amino acid (AA), but it plays an important role in improving feed conversion, promoting growth performance and muscle development, enhancing meat quality in various animal models. Although the general idea of protecting important AAs is not fundamentally novel, but it is very fruitful in practical terms.
It is positive that the following general parameters were measured: average daily gain (ADG); average daily feed intake (ADFI); ADFI/ADG (F/G); crude proteins (CP); ether extract (EE); dry matter (DM); neutral detergent fiber (NDF); acid detergent fiber (ADF) for study of the effect of dietary supplemented with rumen-protected β-alanine (RP-β-Ala) on the growth performance and nutrients apparent digestibility of beef cattle. It is positive that the following serum biochemical and immune parameters of beef cattle were measured: total protein (TP), total cholesterol (TC), triglycerides (TG), blood urea nitrogen (BUN), total antioxidant capacity (T-AOC), superoxide dismutase (SOD), glutathione peroxidase (GSH-Px), malondialdehyde (MDA), tumor necrosis factor-α (TNF-α), interleukin-1β (IL-1β), interleukin-4 (IL-4). It is positive that the ruminal pH value, ruminal ammonia nitrogen; microbial crude protein (MCP); volatile fatty acids (VFA), acetate and propionate were measured. It is interesting that the acetate-to-propionate ratio “tended to decreased in the RP-β-Ala group compared to the control group, while the butyrate concentration did not exhibit significant differences between the two groups”. The rumen microbiota α-diversity of the two groups was evaluated by the Ace, Chao1, Shannon, Simpson, and Coverage indexes, these indexes represent microbial diversity, richness and evenness. The results showed that only Shannon index showed significant change (p < 0.05), indicating certain differences in microbial species richness and evenness between the two groups. The taxa between group differences in abundance at the genus and species level are presented as violin plots to visualize the distribution characteristics of the data. g_Prevotella, g_UCG-004, g_Treponema, g_Selenomonas, g_Moryella, s_Selenomonas_ruminantium_subsp._lactilytica_TAM6421, s_Prevotella_ruminicola, s_Prevotella_sp._BP1-145, s_Selenomonas_ruminantium_AB3002, s_Prevotella_aff._ruminicola_Tc2-24, and s_Treponema_ruminis were significantly enriched in the RP-β-Ala group than in the control group. Contrastingly, g_Mogibacterium, g_Blautia, g_Lachnospiraceae_XPB1014_group, g_DNF00809, g_Desulfovibrio, s_Thermoactinomyces_vulgaris, s_Bacillus_coagulans_g__Bacillus, s_Laceyella_sacchari, and s_Acinetobacter_lwoffii were lower in the RP-β-Ala group than in the control group, etc. It is very positive that the authors presented the general overview of the rumen microbial community and metabolome of the two groups of beef cattle (Figure 7). The performed analysis showed that the dietary supplementation with RP-β-Ala increased the abundance of beneficial bacteria (such as Prevotella, Treponema, and Selenomonas) which improved ruminal fermentation, optimized rumen fermentation capacity, and provided more energy for beef cattle growth (Figure 7). The abundance of harmful bacteria (such as Thermobifida, Thermoactinomyces_vulgaris, and Saccharopolyspora) were inhibited (Figure 7). Moreover, the decreased abundance of Saccharopolyspora led to the suppression of polyamine (spermine and spermidine) biosynthesis. Additionally, the arachidonic acid metabolism pathway was modulated. This suggests that RP-β-Ala could stimulate beef cattle growth by boosting antioxidant capacity and reducing inflammation.
In general, I do not doubt the technical quality of the work and feel that there is a sufficient impact on a broader readership to justify publication in the journal - "Animals". This topic is in the frame of the journal scopes. It is important that the subject matter has been treated in depth. The ethics statements and data availability statements are adequate. The manuscript is relevant for the field, clearly written and presented in a well-structured manner. All the cited references are relevant. This manuscript is scientifically sound and it experimental design is appropriate to test the author’s hypothesis. The manuscript’s results are reproducible based on the details given in the methods section. All the figures and tables are appropriate and properly show the data. The data is interpreted appropriately and consistently throughout the manuscript. The statistical analysis and data acquired are correct. The conclusions consistent with the evidence and arguments presented. Thus, the present manuscript is actual and important in the field of the animal biology and nutrition.
There are no essential comments on the scientific part of this manuscript, but there are some technical remarks. It will be very useful, if the authors can provide small corrections in the text. In particular,
- There are only 15 references (from 50 references in total) that are discussed in the part 1 “Introduction”. It will be very useful, if the authors can describe and discuss more works in the part 1 “Introduction” (from the same 50 references or additional ones that is on the author’s choice).
- It is worth mentioning (in the part 1 “Introduction” or part 4 “Discussion”) that β-alanine serves as one of the elements of the coenzyme A, which in turn is a key activator for metabolites of carbohydrate, lipid and protein metabolism. This can be an additional explanation of the effects of RP-β-Ala on the growth performance and apparent digestibility of nutrients in beef cattle.

Author Response
Response to the reviewer 3
Dear Authors,
the manuscript "Effects of Rumen-Protected β-Alanine on Growth Performance and Microbiome-Host Metabolic Interactions in Beef Cattle" (animals-4001041) by the authors: Daci Fu , Kang Mao , Yihao Zang , Mingren Qu , Qinghua Qiu , Xianghui Zhao , Kehui Ouyang , Yanjiao Li, is related to the important field of animal biology and nutrition. It is aimed as the following: “to investigate the effects of rumen protect β-Alanine (β-Ala) on the growth performance of beef cattle by adding it to the diet, and to explore its specific mechanisms of action by analyzing rumen microbes and their metabolites”. The authors is anticipating that β-Ala addition in ruminant nutrition will provide “a new perspective and scientific basis for the application of β-Ala…”. It is well-known that β-Alanine (3-aminopropionic acid) is a non-essential amino acid (AA), but it plays an important role in improving feed conversion, promoting growth performance and muscle development, enhancing meat quality in various animal models. Although the general idea of protecting important AAs is not fundamentally novel, but it is very fruitful in practical terms.
It is positive that the following general parameters were measured: average daily gain (ADG); average daily feed intake (ADFI); ADFI/ADG (F/G); crude proteins (CP); ether extract (EE); dry matter (DM); neutral detergent fiber (NDF); acid detergent fiber (ADF) for study of the effect of dietary supplemented with rumen-protected β-alanine (RP-β-Ala) on the growth performance and nutrients apparent digestibility of beef cattle. It is positive that the following serum biochemical and immune parameters of beef cattle were measured: total protein (TP), total cholesterol (TC), triglycerides (TG), blood urea nitrogen (BUN), total antioxidant capacity (T-AOC), superoxide dismutase (SOD), glutathione peroxidase (GSH-Px), malondialdehyde (MDA), tumor necrosis factor-α (TNF-α), interleukin-1β (IL-1β), interleukin-4 (IL-4). It is positive that the ruminal pH value, ruminal ammonia nitrogen; microbial crude protein (MCP); volatile fatty acids (VFA), acetate and propionate were measured. It is interesting that the acetate-to-propionate ratio “tended to decreased in the RP-β-Ala group compared to the control group, while the butyrate concentration did not exhibit significant differences between the two groups”. The rumen microbiota α-diversity of the two groups was evaluated by the Ace, Chao1, Shannon, Simpson, and Coverage indexes, these indexes represent microbial diversity, richness and evenness. The results showed that only Shannon index showed significant change (p < 0.05), indicating certain differences in microbial species richness and evenness between the two groups. The taxa between group differences in abundance at the genus and species level are presented as violin plots to visualize the distribution characteristics of the data. g_Prevotella, g_UCG-004, g_Treponema, g_Selenomonas, g_Moryella, s_Selenomonas_ruminantium_subsp._lactilytica_TAM6421, s_Prevotella_ruminicola, s_Prevotella_sp._BP1-145, s_Selenomonas_ruminantium_AB3002, s_Prevotella_aff._ruminicola_Tc2-24, and s_Treponema_ruminis were significantly enriched in the RP-β-Ala group than in the control group. Contrastingly, g_Mogibacterium, g_Blautia, g_Lachnospiraceae_XPB1014_group, g_DNF00809, g_Desulfovibrio, s_Thermoactinomyces_vulgaris, s_Bacillus_coagulans_g__Bacillus, s_Laceyella_sacchari, and s_Acinetobacter_lwoffii were lower in the RP-β-Ala group than in the control group, etc. It is very positive that the authors presented the general overview of the rumen microbial community and metabolome of the two groups of beef cattle (Figure 7). The performed analysis showed that the dietary supplementation with RP-β-Ala increased the abundance of beneficial bacteria (such as Prevotella, Treponema, and Selenomonas) which improved ruminal fermentation, optimized rumen fermentation capacity, and provided more energy for beef cattle growth (Figure 7). The abundance of harmful bacteria (such as Thermobifida, Thermoactinomyces_vulgaris, and Saccharopolyspora) were inhibited (Figure 7). Moreover, the decreased abundance of Saccharopolyspora led to the suppression of polyamine (spermine and spermidine) biosynthesis. Additionally, the arachidonic acid metabolism pathway was modulated. This suggests that RP-β-Ala could stimulate beef cattle growth by boosting antioxidant capacity and reducing inflammation.
In general, I do not doubt the technical quality of the work and feel that there is a sufficient impact on a broader readership to justify publication in the journal - "Animals". This topic is in the frame of the journal scopes. It is important that the subject matter has been treated in depth. The ethics statements and data availability statements are adequate. The manuscript is relevant for the field, clearly written and presented in a well-structured manner. All the cited references are relevant. This manuscript is scientifically sound and it experimental design is appropriate to test the author’s hypothesis. The manuscript’s results are reproducible based on the details given in the methods section. All the figures and tables are appropriate and properly show the data. The data is interpreted appropriately and consistently throughout the manuscript. The statistical analysis and data acquired are correct. The conclusions consistent with the evidence and arguments presented. Thus, the present manuscript is actual and important in the field of the animal biology and nutrition.
Respond: Thank you very much for your thorough review and for your positive assessment of our work. We are delighted to hear that the manuscript is considered scientifically sound, relevant to the readership of Animals, and complete in terms of ethical statements, data availability, and reproducibility. Your confirmation that the experimental design appropriately tests our hypothesis and that the results are interpreted consistently with the evidence gives us confidence that the study meets the journal’s standards. We appreciate your recognition of the depth with which the topic has been treated and will gladly incorporate any specific suggestions you may still have to further improve the clarity or impact of the paper.
There are no essential comments on the scientific part of this manuscript, but there are some technical remarks. It will be very useful, if the authors can provide small corrections in the text. In particular
Comment 1 There are only 15 references (from 50 references in total) that are discussed in the part 1 “Introduction”. It will be very useful, if the authors can describe and discuss more works in the part 1 “Introduction” (from the same 50 references or additional ones that is on the author’s choice).
Respond: Thank you for this valuable suggestion. We have supplemented relevant references in the introduction:
8 Song P.; Zhang X.; Wang S.; Xu W.; Wei F. Advances in the synthesis of β-alanine Front Bioeng Biotechnol. 2023 undefined, 11, pp1283129
9 Ong SW, Chen WL, Chien KY, Hsu CW. Dosing strategies for β-alanine supplementation in strength and power performance: a systematic review. J Int Soc Sports Nutr. 2025, 22, pp2566368.
14 Ahn DH.; Ko YS.; Prabowo CPS.; Lee SY. Microbial production of propionic acid through a novel β-alanine route. Metab Eng. 2026, 93, pp219-231
Comment 2 It is worth mentioning (in the part 1 “Introduction” or part 4 “Discussion”) that β-alanine serves as one of the elements of the coenzyme A, which in turn is a key activator for metabolites of carbohydrate, lipid and protein metabolism. This can be an additional explanation of the effects of RP-β-Ala on the growth performance and apparent digestibility of nutrients in beef cattle.
Respond: Thank you for this valuable suggestion. We have now added the following sentences in Lines 71–75 of the revised Introduction:
"Apart from being a carnosine precursor, β-Ala is also a constituent of pantothenic acid (vitamin B₅), which is an essential component of coenzyme A (CoA). CoA acts as the universal carrier of acyl groups, thereby directly participating in the catabolic and anabolic pathways of carbohydrates, lipids, and amino acids"

Round 2
Reviewer 1 Report
Comments and Suggestions for Authors
Overall, the authors have addressed the major concerns and the revised manuscript is much improved. However, I recommend a Minor Revision to correct a few remaining, mostly editorial/technical issues before final acceptance.
Specific comments
Table 1: The term "straw silage" is ambiguous — "straw" does not identify a specific crop. Please clarify the botanical/source material (e.g., wheat straw silage, rice straw silage, barley straw silage).
Line 150: please capitalize the phrase — change "standard laboratory sieve" to "Standard laboratory sieve." Also check capitalization consistency for instrument and equipment names throughout the Methods.
Line 208: Please add a formal citation for the pheatmap R package (and likewise ensure ggpubr is cited formally). R packages have authors/maintainers and versions — use citation("pheatmap") and citation("ggpubr") in an R session to obtain the exact author names, year and recommended citation format, and include the package version (packageVersion("pheatmap")).
Line 214: The Methods state that results are presented as mean ± SEM, but I could not find mean and SEM values reported in any table.
Figure 1: Digestibility should be presented on a 0–100% scale and the y‑axis labeled accordingly (e.g., "Apparent digestibility (%)") rather than 0–1. If the authors prefer to keep proportions, they must at minimum standardize the number of decimal places across panels — currently values are inconsistently shown with 1–3 decimal places. In addition.. Figure 1 description still contain underscores. Please remove all underscores from figure text and ensure consistency across the manuscript.

Author Response
Thank you very much for your positive feedback and for confirming that our revisions have adequately addressed all key concerns. We appreciate your recognition of the improved scientific rigor and clarity.

Reviewer 2 Report
Comments and Suggestions for Authors
The authors' responses to the various points raised are comprehensive and satisfactory, and all key concerns have been adequately addressed. The revised manuscript demonstrates a marked improvement in scientific rigor, methodological soundness, and clarity, and now meets the journal's publication standards.
Author Response

(The authors gave the same response as above.)
